# Preliminary Studies on the Use of Reactive Distillation in the Production of Beverage Spirits

**Jacob D. Rochte [1,]*** and **Kris A. Berglund [1,2]**

1   Department of Chemical Engineering and Material Science, Michigan State University,
    East Lansing, MI 48823, USA; berglund@msu.edu
2   Luleå Biochemical Process Engineering, University of Technology, SE-97187 Luleå, Sweden
*   Correspondence: rochteja@msu.edu; Tel.: +1-(517)-802-7095

**Abstract:** Distilled alcoholic beverages have been produced through fermentation and distillation for centuries but have not purposefully involved a chemical reaction to produce a flavoring. Introducing a microorganism to produce butyric acid along with the typical yeast ethanol fermentation sets up a reactive distillation system to flavor a spirit with ethyl butyrate and butyric acid. The ternary interactions of water, ethanol, and butyric acid allow all three to vaporize in the stripping distillation, thus they are concentrated in the low wines and give a large excess of ethanol compared to butyric acid for better reaction completion. The stripping distillation has also been modeled on Aspen Plus® V9 software (by Aspen Technology, Inc. Bedford, MA, USA) and coincides well with a test stripping distillation at the bench scale. Amberlyst® 15 wet catalyst was added to a subsequent distillation, resulting in the production of the desired ethyl butyrate in the distillate, measured by gas chromatography. Primary sensory evaluation has determined that this process has a profound effect on the smell of the spirit with the main flavor being similar to fruity bubble gum. The current results will provide a pathway for creating spirits with a desired flavor on demand without acquiring a heavy capital cost if a beverage distillation column is already purchased.

**Keywords:** reactive distillation; esterification; spirits; ethyl butyrate; butyric acid; ethanol

## 1. Introduction

The basic concept of fermentation and distillation to create an alcoholic beverage has remained largely unchanged since its inception, mainly due to tradition. There have been advancements made as knowledge of the process grows. For instance, it is now known why the use of copper is important in a still, as it has a catalytic effect of removing sulfur compounds during distillation. Studies have determined that the main compound being removed by the copper is dimethyl trisulfide which has a low odor detection threshold (33 parts per trillion in 20% ethanol) [1].

Reactive distillation combines the two unit operations of a chemical reaction and a distillation into one [2–4]. This technique is useful for reactions that need a special condition that can be achieved through distillation, such as a large excess of one of the reactants, or for removing a product to drive an equilibrium reaction to completion [2,4,5].

Combining two processes into one decreases time and energy to produce a desired result. Initial capital cost and maintenance costs are also lowered because the reactor and column is the same piece of equipment. A common use for reactive distillation is esterification. Other than sulfur adsorption onto copper, the alcoholic beverage industry does not purposefully use this technique in the production of spirits.

Introducing esterification into the production of beverages' spirits will create a food grade, flavored spirit while adhering to the standards of identity for distilled alcoholic

beverages in the United States (27 CFR Section 5.22 https://www.ecfr.gov/cgi-bin/text-idx?
SID=39303a4811a9732df1b66270946fee1d&mc=true&node=pt27.1.5&rgn=div5#se27.
1.5_122). This paper will discuss the materials and unit operations that will be combined to develop
this process and create the proposed spirit.

*Esters*

An ester is formed when an acid and alkoxy group join together though a condensation reaction.
It can be any acid and any alkoxy group, but is usually an organic acid and an alcohol. As this is a slow
reaction without the presence of a catalyst, a classic method is to use the Fischer-Speier esterification
with a sulfuric acid catalyst. Figure 1 below shows the condensation of butyric acid and ethanol to
create ethyl butyrate; this is the reaction that is desired in this beverage research.

**Figure 1.** The chemical equation for the dehydration reaction of ethanol and butyric acid to form ethyl
butyrate and water.

Esters are commonly used in flavorings and fragrances because of their pleasant odor. Ethyl
butyrate is commonly used because it has the smell of pineapples or fresh orange juice [6].

## 2. Modeling

Modeling techniques were used to determine if this study was possible. A ternary residue curve
map (RCM) was made with the three main compounds at the corners: water, ethanol, and butyric acid.
Aspen Plus$^{®}$ was also used to design a column that shows this system would work for the stripping
portion of the beverage distillation process.

### 2.1. Ternary Residue Curve Map

Figure 2 shows a ternary RCM of water, ethanol, and butyric acid in weight fraction. Along with
the ethanol/water azeotrope, a binary azeotrope is present between water and butyric acid which boils
at 99.5 °C. A separatrix (or simple batch distillation boundary) is drawn in green which divides the
diagram into two distillation regions. Residue curves are drawn in light blue with arrows indicating
increasing temperature in the pot (or time during the distillation).

The concentrations of water, ethanol, and butyric acid in the pot at the beginning of a run
determine the starting point on Figure 2 [2,7]. The residue curves show the concentration of the
components in the pot as the distillation proceeds. As an example, if the concentrations of water,
ethanol, and butyric acid are 75% *w/w*, 20% *w/w*, and 5% *w/w* respectively, the pot concentration will
be 100 % *w/w* water when the distillation is finished with all of the ethanol and butyric acid from the
system in the distillate. If the starting point was to the left of the distillation boundary, the pot would
eventually consist of pure butyric acid.

The presence of these binary azeotropes in this system makes it possible for the high boiling
butyric acid to be carried over in the distillate stream by the water during the first distillation of the
spirit. This then sets up the system for the second distillation over a catalyst to produce the desired
ethyl butyrate in the finished spirit.

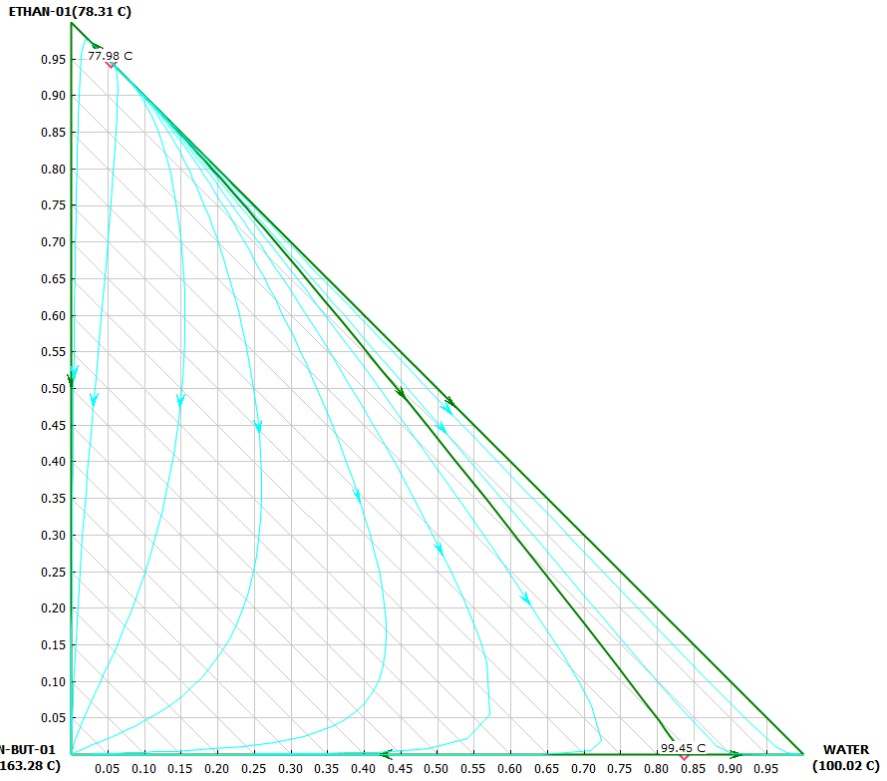

**Figure 2.** A ternary residue curve map of water, ethanol, and butyric acid with separatrix. The separatrix is drawn in green which starts at the ethanol/water azeotrope and ends at the butyric acid/water azeotrope. Residue curves are drawn in light blue. This diagram is measured in weight fraction.

## 2.2. Aspen Plus® Column Design

A column was designed in Aspen Plus® to distill a whiskey mash with a requirement to have butyric acid present in the distillate of the column. The column was designed to have 17 bubble cap trays, stage seven as the feed stage, and a reflux ratio of five. The following Figures 3–6 are screenshots from the Aspen Plus® software used for the simulation.

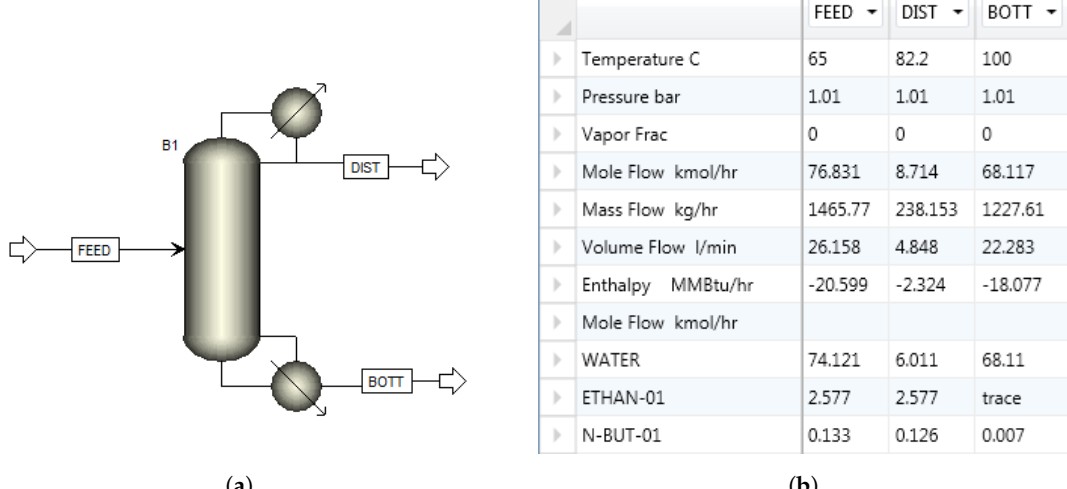

|  | FEED | DIST | BOTT |
|---|---|---|---|
| Temperature C | 65 | 82.2 | 100 |
| Pressure bar | 1.01 | 1.01 | 1.01 |
| Vapor Frac | 0 | 0 | 0 |
| Mole Flow kmol/hr | 76.831 | 8.714 | 68.117 |
| Mass Flow kg/hr | 1465.77 | 238.153 | 1227.61 |
| Volume Flow l/min | 26.158 | 4.848 | 22.283 |
| Enthalpy MMBtu/hr | -20.599 | -2.324 | -18.077 |
| Mole Flow kmol/hr |  |  |  |
| WATER | 74.121 | 6.011 | 68.11 |
| ETHAN-01 | 2.577 | 2.577 | trace |
| N-BUT-01 | 0.133 | 0.126 | 0.007 |

(**a**)    (**b**)

**Figure 3.** Aspen Plus® screenshots of the flowsheet and overall stream table. (**a**) Flowsheet. (**b**) Stream Table.

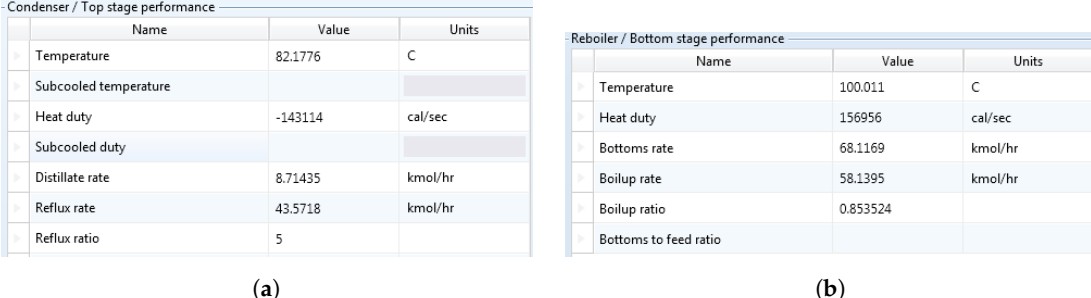

(**a**)　　　　　　　　　　　　　　　　(**b**)

**Figure 4.** Aspen Plus® screenshots of the performance of the top and bottom stages. (**a**) Top Stage Performance. (**b**) Bottom Stage Performance.

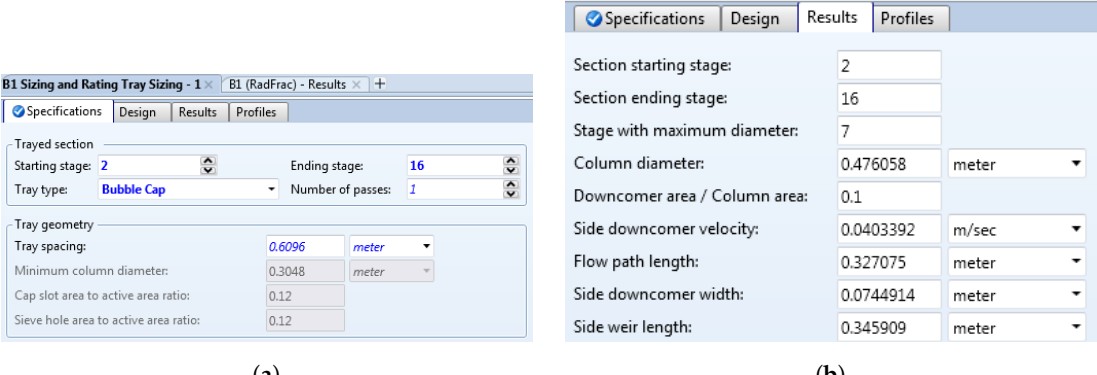

(**a**)　　　　　　　　　　　　　　　　(**b**)

**Figure 5.** Aspen Plus® screenshots for the sizing of the column. (**a**) Tray Sizing Input. (**b**) Tray Sizing Results.

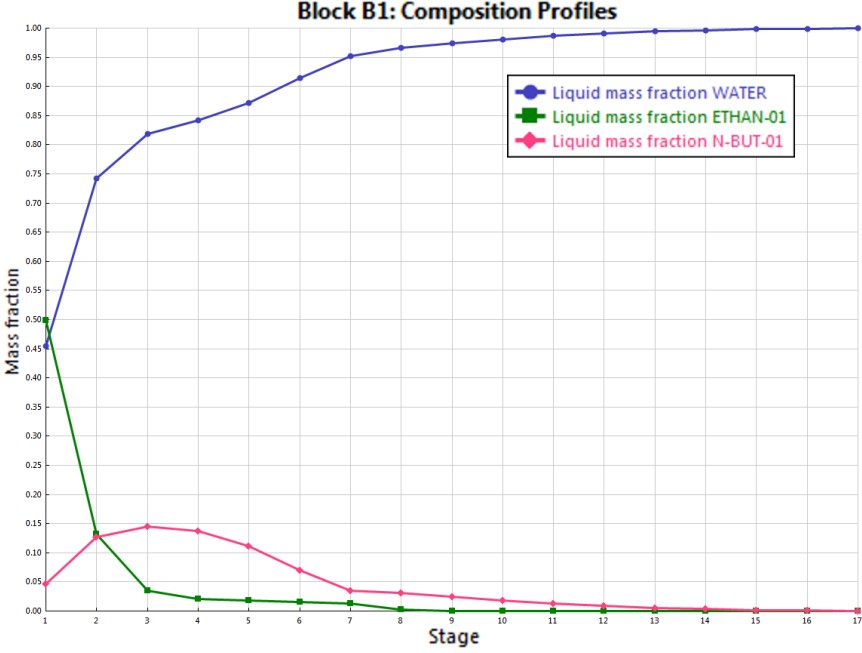

**Figure 6.** A graph of the column tray compositions as produced by Aspen Plus® .

As seen in Figure 3b, the distillate stream of the column contains 94.7 mol % of the butyric acid in the system. This simulation shows that the main components will be present in the distillate of the stripping run. The low wines produced from this can then be distilled again using a catalyst to convert the butyric acid into the desired ethyl butyrate.

## 2.3. Lab-scale Fractional Distillation

A lab fractional distillation was performed in conjunction with the Aspen Plus® simulation to support the results. The experiment used a 500 mL round-bottom flask, Vigreux column, and a straight condenser setup in a fractional distillation configuration. The flask started with 250 mL of an aqueous solution containing 79.2 g/L ethanol and 4 g/L butyric acid. Figure 7 shows the concentrations of ethanol and butyric acid of the experiment as a function of distillate volume collected. This ternary distillation was performed to determine at what point the butyric acid is present in the distillate.

A final distillate of 70 mL was collected containing 230 g/L ethanol and 6.2 g/L butyric acid. This experiment coincides well with the Aspen Plus® simulation in Section 2.2.

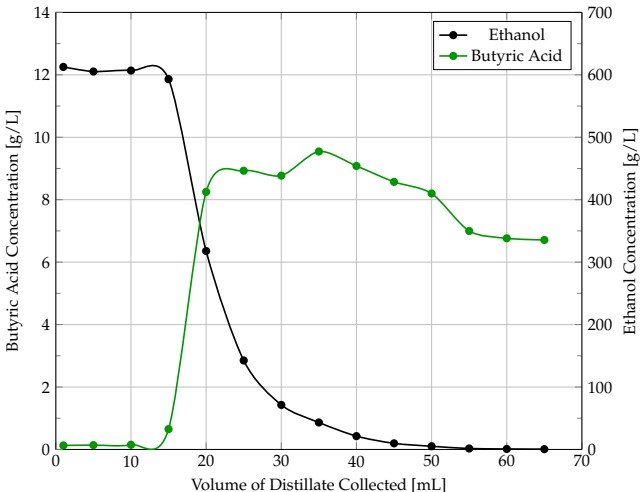

**Figure 7.** Distillate concentrations of ethanol and butyric acid during the lab-scale fractional distillation vs. the volume of distillate collected.

## 3. Materials and Methods

This research was preformed on the bench scale. The mash was stripped 1 L at a time in a round-bottom flask. The flask was connected to a 36 cm vigreux column, a 75° distillation adapter, and finally a 20 cm straight Liebig condenser. All connections are ⊤ 24/40. Six strips of 14 AWG copper wire (1.63 mm diameter) were put into the vigreux column to represent the copper in a beverage distillation column.

For each stripping run, the distillate was collected until the ethanol fell below 10% ABV. These low wines were then distilled a second time over Amberlyst® 15 wet catalyst with varying loadings and positions within the column. All samples were analyzed on a Shimadzu GC-17A with FID. The column used was a Zebron ZB-WAXplus 30 m × 0.25 mm × 0.25 μm. The GC oven was ramped up from 40 °C to 140 °C to analyze everything including butyric acid. The column needed to be shortly reconditioned by quickly ramping up to 220 °C at the end of every run.

## 4. Results

### 4.1. Reactive Distillation

Figure 8 shows component concentrations in a reactive spirit distillation as a function of distillate volume collected. The distillation consisted of 150 mL of low wine diluted to 30% ABV. Butyric acid was added to the pot to give a 4.0 g/L starting concentration and 15 g of Amberlyst® 15 wet catalyst was contained in a mesh bag and added to the pot during the distillation; no ethyl butyrate was added prior to the experiment, as seen in Table 1.

**Table 1.** Compounds of interest in the reactive distillation system. Concentrations in [g/L].

| Sample | Ethanol | Ethyl Butyrate | Butyric Acid |
|---|---|---|---|
| Initial Pot | 240.81 | 0.00 | 4.02 |
| Final Pot | 0.00 | 0.00 | 1.64 |
| Total Distillate | 349.31 | 1.29 | 4.82 |

**Figure 8.** Component concentrations in the distillate of a spirit run with 150 mL of low wines at 30% ABV, 4.0 g of butyric acid, and 15 g of Amberlyst® 15 wet catalyst in the pot.

As seen in Figure 8, the ethanol and ethyl butyrate concentration curves have the same shape and are inverse of the butyric acid concentration curve. During the distillation, butyric acid was reacted with ethanol over the catalyst to form the desired ethyl butyrate. This product was then carried over in the distillate stream by the ethanol.

Note: If creating a spirit, the heads would have been cut after collecting 5 mL, and the tails cut after collecting 40 mL. These cuts would have resulted in the final hearts product being 35 mL in volume and having low ethyl acetate and butyric acid concentrations.

*4.2. Sensory Evaluation*

Four variations of the reactive distillation from Section 4.1 were distilled and 35 mL of hearts were collected from each as discussed above. Table 2 shows the differences in the four samples.

**Table 2.** Difference in the samples used for the sensory evaluation to determine if this process has an effect on the final spirit.

| Sample | Butyric Acid [g/L] | Catalyst [g] |
|---|---|---|
| 1 | 0 | 0 |
| 2 | 0 | 7.5 |
| 3 | 5 | 0 |
| 4 | 5 | 7.5 |

Each sample was proofed to 40% ABV with deionized water before being analyzed. The participant was asked to smell each sample and give a description of what they smelled (Table 3). They were also asked if they could tell a difference between samples (Table 4).

**Table 3.** Sample notes from individual sensory evaluations.

| Participant | Sample 1 | Sample 2 | Sample 3 | Sample 4 |
|---|---|---|---|---|
| 1 | whiskey | softer than 1 | N/A | juicy fruit gum |
| 2 | corn | corn | cheese | licorice/fruity |
| 3 | alcohol | alcohol | same as 2 | apples |
| 4 | corn | corn | corn | bubble gum |
| 5 | corn whiskey | corn whiskey | buttered popcorn | buttered popcorn |
| 6 | Popsicle stick | less strong, but same as 1 | Olay bar soap | minty |
| 7 | whiskey | bubble yum | soap | sour candy, apple |
| 8 | plastic, organic, sweet | less sweet | N/A | popcorn |

**Table 4.** Could the participant tell the difference between samples of the sensory evaluation.

| Participant | Sample | | | | | |
|---|---|---|---|---|---|---|
| | 1v2 | 1v3 | 2v3 | 1v4 | 2v4 | 3v4 |
| 1 | yes | no | yes | yes | yes | yes |
| 2 | no | yes | yes | yes | yes | yes |
| 3 | yes | yes | no | yes | yes | yes |
| 4 | yes | yes | yes | yes | yes | yes |
| 5 | no | yes | yes | yes | yes | yes |
| 6 | yes | yes | yes | yes | yes | yes |
| 7 | yes | yes | yes | yes | yes | yes |
| 8 | yes | no | yes | yes | yes | yes |

The evaluation showed that samples 1 and 2 were similar with a smell of whiskey and they could be told apart; sample 1 had a stronger whiskey smell than sample 2. Sample 3 had notes of licorice and popcorn, which is expected as there was butyric acid present. Sample 4 was overwhelmingly different than the others having a fruity smell similar to that of banana candy or bubble gum.

## 5. Discussion

For research purposes, food grade butyric acid was added to the fermented mash before the first distillation. Future work will develop a co-fermentation between *Saccharomyces cerevisiae* and *Clostridium tyrobutyricum* to produce ethanol and butyric acid, respectively.

Introduction of ester production in beverage spirit distillations will create a new series of products that have unique smells and tastes. These qualities will give the spirit an increased value and marketability, thus mitigating the reduced ethanol yield from the production of the organic acid in the fermentation.

This research will provide a pathway for creating spirits with a desired flavor on demand without acquiring a heavy capital cost if a beverage distillation column is already purchased. Manufacturers would be able to produce a uniquely flavored spirit of their own with only the purchase of the correct catalyst and microorganism.

The code of federal regulations (CFR) has set rules of what can and cannot be done when it comes to distilled spirits, specifically (27 CFR Section 5 https://www.ecfr.gov/cgi-bin/text-idx?SID= 39303a4811a9732df1b66270946fee1d&mc=true&node=pt27.1.5&rgn=div5).

The reaction of ethanol and butyric acid to ethyl butyrate proceeds naturally. The catalyst that is put into the column speeds the reaction rate and drives the reaction more toward the product, i.e., the catalyst is not adding a new reaction, only accelerating an already existing reaction.

**Author Contributions:** J.D.R. wrote this paper in preparation of a doctoral dissertation under advisement from K.A.B.

**Funding:** This research received no external funding.

**Acknowledgments:** Thank you to those individuals who took part in the sensory evaluation study.

**Conflicts of Interest:** The authors declare no conflict of interest. The founding sponsors had no role in the design of the study; in the collection, analyses, or interpretation of data; in the writing of the manuscript, and in the decision to publish the result.

## Abbreviations

The following abbreviations are used in this manuscript:

RCM    Residue Curve Map
g/L    grams per liter
GC    Gas Chromatograph
AWG    American Wire Gauge
CFR    Code of Federal Regulations
ABV    Alcohol By Volume

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
