# Peer review of "Preliminary Studies on the Use of Reactive Distillation in the Production of Beverage Spirits"

_beverages, doi:10.3390/beverages5020029_

Reviewer 1 Report

1.                  The section „Introduction” - There is too little information about other authors results regarding this method of distillation. The authors should also more strongly justify the application character of this method.

2.                  Page 2, line 38 – please change „alkoxy” to „alkoxy group”.

3.                  Page 5, lines 76 and 81 – „preformed”?

4.                  Page 5, line 79 – the term „grams per litre” is it unnecessary.

5.                  Page 7, figure 8 – the term „iso-Amyl” should by replace with „iso-Amyl alcohol”.

6.                  Section „Materials and methods” - Why the authors give results in the "Materials and methods" section? Please combine Results in section „Results and discussion”

7.                  The section 3.2 sensory evaluation lacks detailed information about the sensory method (how many panellists were involved, which descriptor was assessment? – smell, taste?). What was the strength of heart factions being assessed? Where is the table or diagram with the sensory evaluation results? 

8.                  The section „Discussion” lacks real discussion with the results of other authors.

Author Response

1.       The section „Introduction” - There is too little information about other authors results regarding this method of distillation. The authors should also more strongly justify the application character of this method.

·         There were no other authors with similar work. This work resulted in a provisional patent that is the process of being submitted as a full patent.

2.       Page 2, line 38 – please change „alkoxy” to „alkoxy group”.

·         done

3.       Page 5, lines 76 and 81 – „preformed”?

·         done

4.       Page 5, line 79 – the term „grams per litre” is it unnecessary.

·         done

5.       Page 7, figure 8 – the term „iso-Amyl” should by replace with „iso-Amyl alcohol”.

·         done

6.       Section „Materials and methods” - Why the authors give results in the "Materials and methods" section? Please combine Results in section „Results and discussion”

·         Results section added after “Material and Methods” and “Discussion”

7.       The section 3.2 sensory evaluation lacks detailed information about the sensory method (how many panellists were involved, which descriptor was assessment? – smell, taste?). What was the strength of heart factions being assessed? Where is the table or diagram with the sensory evaluation results? 

·         The sensory evaluation was defined more and tables added to show the results per each participant.

8.       The section „Discussion” lacks real discussion with the results of other authors.

·         Similar results could not be found

Reviewer 2 Report

is manuscript is well written and its application is of interest.The only criticism I have that there is very little background information included in the manuscript. I would urge the authors to ex This version is acceptable. Please accept in its current form.

Author Response

Thank you for your review. There is a revised version per the other reviewer.